# Concussion-Related Biomarker Variations in Retired Rugby Players and Implications for Neurodegenerative Disease Risk: The UK Rugby Health Study

**DOI:** 10.3390/ijms25147811

**Published:** 2024-07-17

**Authors:** Norah Alanazi, Melinda Fitzgerald, Patria Hume, Sarah Hellewell, Alex Horncastle, Chidozie Anyaegbu, Melissa G. Papini, Natasha Hargreaves, Michal Halicki, Ian Entwistle, Karen Hind, Paul Chazot

**Affiliations:** 1Department of Biosciences, Wolfson Research Institute for Health and Wellbeing, Durham University, Durham DH1 3LE, UK; norah.alanazi@durham.ac.uk (N.A.); alex.horncastle@durham.ac.uk (A.H.); natasha.hargreaves@durham.ac.uk (N.H.); michal.halicki@balliol.ox.ac.uk (M.H.); ian.entwistle@durham.ac.uk (I.E.); 2Curtin Health Innovation Research Institute, Curtin University, Bentley, WA 6102, Australia; lindy.fitzgerald@curtin.edu.au (M.F.); sarah.hellewell@curtin.edu.au (S.H.); melissa.papini@curtin.edu.au (M.G.P.); 3Perron Institute for Neurological and Translational Sciences, Nedlands, WA 6009, Australia; 4Sports Performance Research Institute New Zealand (SPRINZ), Faculty of Health and Environmental Science, Auckland University of Technology, AUT Millennium, 17 Antares Place, Mairangi Bay, Private Bag 92006, Auckland 1142, New Zealand; patria.hume@aut.ac.nz; 5Traumatic Brain Injury Network (TBIN), Auckland University of Technology, Private Bag 92006, Auckland 1142, New Zealand; 6Auckland Bioengineering Institute, The University of Auckland, Private Bag 92019, Auckland 1142, New Zealand; 7Technology and Policy Laboratory, The University of Western Australia, 35 Stirling Highway, Perth, WA 6009, Australia; 8Centre for Neuromuscular & Neurological Disorders, University of Western Australia, Crawley, WA 6009, Australia; 9Wolfson Research Institute for Health and Wellbeing, Durham University, Durham TS17 6BH, UK

**Keywords:** concussion, brain injury, sport, neurodegenerative disease, biomarkers

## Abstract

The health and well-being of retired rugby union and league players, particularly regarding the long-term effects of concussions, are of major concern. Concussion has been identified as a major risk factor for neurodegenerative diseases, such as Alzheimer’s and Amyotrophic Lateral Sclerosis (ALS), in athletes engaged in contact sports. This study aimed to assess differences in specific biomarkers between UK-based retired rugby players with a history of concussion and a non-contact sports group, focusing on biomarkers associated with Alzheimer’s, ALS, and CTE. We randomly selected a sample of male retired rugby or non-contact sport athletes (n = 56). The mean age was 41.84 ± 6.44, and the mean years since retirement from the sport was 7.76 ± 6.69 for participants with a history of substantial concussions (>5 concussions in their career) (n = 30). The mean age was 45.75 ± 11.52, and the mean years since retirement was 6.75 ± 4.64 for the healthy controls (n = 26). Serum biomarkers (t-tau, RBP-4, SAA, Nf-L, and retinol), plasma cytokines, and biomarkers associated with serum-derived exosomes (Aβ42, p-tau181, p-tau217, and p-tau231) were analyzed using validated commercial ELISA assays. The results of the selected biomarkers were compared between the two groups. Biomarkers including t-tau and p-tau181 were significantly elevated in the history of the substantial concussion group compared to the non-contact sports group (t-tau: *p* < 0.01; p-tau181: *p* < 0.05). Although between-group differences in p-tau217, p-tau231, SAA, Nf-L, retinol, and Aβ42 were not significantly different, there was a trend for higher levels of Aβ42, p-tau217, and p-tau231 in the concussed group. Interestingly, the serum-derived exosome sizes were significantly larger (*p* < 0.01), and serum RBP-4 levels were significantly reduced (*p* < 0.05) in the highly concussed group. These findings indicate that retired athletes with a history of multiple concussions during their careers have altered serum measurements of exosome size, t-tau, p-tau181, and RBP-4. These biomarkers should be explored further for the prediction of future neurodegenerative outcomes, including ALS, in those with a history of concussion.

## 1. Introduction

Rugby, a widely played contact sport, is associated with a high risk of injury, including concussions, reflecting its physical nature involving multiple and repetitive tackling and collisions [1]. Concussions are brain injuries produced by accelerations and decelerations of the head and can be characterized by a sudden brief impairment of consciousness, paralysis of a reflex activity, and/or loss of memory [2]. Common signs and symptoms of concussion are headache, dizziness, sleep disturbance, balance disturbances, confusion/disorientation, cognitive problems, and emotional difficulties [3]. Concussions are concerning given their link to long-term cognitive, motor, and mental health effects [4,5,6].

Studies also indicate that the accumulating effects of recurrent concussions can have long-term consequences with the risk of subsequent brain injury and neurological health. Tsushima et al. [7] found that athletes with a prior concussion had a 3–5 times greater risk of sustaining a subsequent concussion than those with no history of a concussion. Furthermore, a history of concussions could greatly affect the cognitive domains of memory, executive function, and psychomotor function. Retired athletes in general sports with a history of concussions have increased cognitive difficulties [8]. Kerr et al. [9] studied 204 (84.3% with a concussion history) former collegiate football players, aiming to examine the associations between concussion history and adverse health outcomes. The authors demonstrated a link between concussion history and health outcomes, such as general physical and mental health, depression, and alcohol dependence.

The concern is accelerating regarding the health and wellness of former participants in contact sports, as well as the possible long-term repercussions of concussions and other brain impairments [1,6]. A significantly higher frequency of amyotrophic lateral sclerosis (ALS) has been reported in former professional and varsity soccer players and American football players [10]. In a cohort mortality study of 3439 National Football League (NFL) players to analyze neurodegenerative causes of death, Lehman et al. [11] concluded that neurodegenerative-related mortality of NFL players, resulting from conditions that include Alzheimer’s and ALS, was three times higher than in the general population [11]. In an earlier study, Guskiewicz et al. [12] administered a general health questionnaire to 2552 retired professional football players (61% with at least one concussion and 24% with three or more concussions throughout their professional football career). The results indicated a higher risk for the onset of dementia-related syndromes in professional football players who reported a higher number of repetitive concussions [12]. Prior studies also support a link between traumatic brain injury (TBI) and increased risk of cognitive impairment and dementias such as Alzheimer’s disease [4]. A retrospective cohort study [13] found that mortality from neurodegenerative disease was around three times higher in the 7676 former Scottish professional soccer players compared with 23,028 age-matched controls from the general population. While the link between TBI, repetitive concussions, and neurodegenerative risk has been widely reported [14,15,16,17], research is needed to determine the neurobiological relationships between concussions, persistent negative behavioral effects [6], markers of disease onset and progression, and targets for therapy [18].

In a recent review study of TBI and the risk of neurodegenerative disorder [19], TBI of any severity was associated with a 63–96% increased risk of all-cause dementia. Given that there is no current cure for dementia, early detection of this disease is crucial to support the management and monitoring of the disease trajectory. Unfortunately, many neurodegenerative illnesses that underlie dementia cannot be diagnosed or treated in the prodromal, or early stages, because there are no clinically accepted biomarkers for these diseases [20]. Furthermore, no study has yet investigated the potential concussion biomarkers that could be linked to neurodegenerative diseases in former athletes with a history of sports-induced concussions.

The current study aimed to explore biomarkers indicative of neurodegeneration in a well/defined sample of retired rugby players years after the end of their career with a history of multiple concussions (chronic state), relative to healthy controls. The biomarkers included were total tau (t-tau), three forms of phosphorylated tau (p-tau), serum amyloid Alpha (SAA), beta-amyloid (Aβ42), retinol-binding protein 4 (RBP-4), retinol, and neurofilament light (Nf-L), and exosome size. Previous studies have explored t-tau as a potential biomarker of concussion/TBI [21], and t-tau measured in cerebrospinal fluid (CSF) has been associated with neuronal and axonal degeneration [22]. Moreover, a recent study reported associations between the number of concussions with levels of t-tau in CSF in former National Football League players [23].

Plasma p-tau181 has been suggested as a promising blood-based biomarker [24], that could anticipate Alzheimer’s [25]. Devoto et al. [26] reported a significant increase in p-tau181 plasma concentrations from acutely concussed collegiate athletes and mild TBI groups when compared to non-concussed controls. Increased p-tau181 plasma concentrations were also reported in a similar study [27]. Another potential biomarker for Alzheimer’s is p-tau217 [28], with concentrations accumulating in the brain as AD develops [28]. P-tau217 can distinguish Alzheimer’s from other neurological disorders with a higher degree of accuracy compared to recognized MRI- and plasma-based biomarkers [29]. Turk et al. [30] found that levels of p-tau231 were significantly higher in low and high CTE groups compared to the healthy control group and the AD group. The authors suggested that p-tau231 levels could be more sensitive and specific than p-tau181 in differentiating CTE from non-CTE and non-Alzheimer’s disease, and for differentiating CTE from Alzheimer’s disease. Furthermore, a post-mortem study of 473 cadavers with neuropathologically diagnosed AD/CTE [31] demonstrated that those with CTE had significantly higher p-tau231 but not p-tau181 compared to controls [31].

SAA protein has been associated with processes related to inflammation, pathogen defense, HDL metabolism, and cholesterol transport and plays an important role in atherosclerosis, rheumatoid arthritis, Alzheimer’s disease, and cancer [32]. It is an acute-phase protein upregulated inside the liver after TBI [33], produced in the brain, and co-localized with senile plaques in patients with Alzheimer’s [34]. Also, Aβ42 was chosen as a focus for this current study as a reliable biomarker of Alzheimer’s [35]. Lejbman et al. [36] reported in a study of 172 military personnel a significant increase in Aβ40 and a tendency to elevate concentrations in Aβ42 in patients with TBI history.

Nf-L is currently among the most promising biomarker candidates being researched [37]. Neurofilaments are major components of the axonal cytoskeleton and consist of three types of protein chains: a light chain, an intermediate chain, and a heavy chain [37]. In neurodegenerative diseases, axonal damage may lead to the release of neurofilaments into the CSF and blood [37]. It was found that plasma Nf-L concentrations were elevated in military personnel who had sustained repetitive TBIs compared with those with one or two TBIs long-term [38]. Furthermore, serum levels of Nf-L were higher in amateur boxers compared to a baseline, with a correlation between the level of increase and the number of head blows suffered by the boxers [39].

Retinol, retinal, and retinoic acid (RA) are all referred to as vitamin A [40]. RA is essential for the development of the nervous system in embryos, especially in the early stages of pregnancy. Both excess and lack of RA can lead to teratogenic neural tube abnormalities [41]. RA metabolites remain engaged in the adult brain’s neuronal differentiation, axonal outgrowth, myelination and remyelination, and blood–brain barrier integrity, suggesting a potential role for RA following brain injury [42]. It is well reported that the retinoid system is compromised in ageing and Alzheimer’s disease.

RBP-4 is an adipocyte-secreted hormone (adipokine) that regulates insulin signaling and is also a key transporter of retinoic acid and its derivatives [43]. RBP-4 is strongly expressed in areas of the brain important for cognitive processing [44]. RBP4 can also bind to transthyretin, which can act as a carrier protein that is believed to modulate Aβ levels by transporting Aβ from the brain to the periphery resulting in lower amounts of toxic Aβ in the brain [43]. Therefore, RBP-4 may be involved in pathways related to neurodegenerative pathobiologies, such as Alzheimer’s and ALS. Serum levels of RBP-4 have been shown to be dramatically reduced following TBI, suggesting that the serum level of this protein is a useful biomarker for patients who are brain-injured [44].

Cytokines have been linked to acute TBI/concussion [45,46,47]. Nitta et al. [45] investigated serum inflammatory markers that could predict symptom recovery after sport-related concussion (SRC) among 41 athletes with concussion and 43 controls. IL-6 was significantly increased in serum following TBI, suggesting that it can be used as a serum biomarker of SRC and to identify athletes at risk for prolonged recovery following SRC [45]. A study of 94 military personnel participants who sustained concussions and controls revealed a significant elevation of IL-6 concentration levels in the concussed military personnel group less than 8 h following the injury [46]. Sun et al. [47] investigated two cohorts of individuals within 1 week of mTBI and a healthy control. They observed that serum levels of IL-1β, IL-6, and CCL2 were acutely increased in patients with mTBI compared to controls and that serum cytokine levels can be used for evaluating post-concussion symptoms and predicting cognitive outcomes in patients with mTBI [47]. However, serum cytokine levels in the chronic phase after single or repeated mTBI are not well understood.

Exosomes are extracellular vehicles (EVs) produced in the endosomes of eukaryotic cells [48] and present in most body fluids, including saliva, plasma, and breast milk [49]. They are defined specifically by their diameter (~30–150 nm) [50]. Exosomes are considered crucial vehicles for intercellular communication as they carry functionally active messenger RNAs (mRNAs), microRNAs (miRNAs), proteins, and lipids between cells to mediate a range of biological effects upon target cell binding and uptake [51]. The importance of exosomes in the pathogenesis of neurodegeneration in vivo has not yet been established, but there is a suggestion that they represent cellular damage [52]. Therefore, this study will examine the sizes of these particles and the levels of biomarkers associated with exosomes in the chronic phase following retirement from sport.

## 2. Results

A total random selection of 56 males (with a history of concussions n = 30 and control n = 26) participated in this study. Participants’ information is given in Table 1 below.

### 2.1. Serum Biomarker Levels in Concussed and Control Groups

A series of related serum biomarkers were explored (Figure 1). There was a highly significant elevation of serum t-tau concentration in the serum of a sub-set of retired rugby players who had suffered multiple concussions when compared to the non-contact control group (** *p* < 0.01). Concentrations of serum RBP-4 were significantly lower in the retired rugby group compared to the healthy control group (* *p* < 0.05). Unexpectedly, our results showed that there was no difference between the levels of SAA in serum between the two groups. There were also no differences in Nf-L and retinol between the concussed and control groups.

A series of chemokines and cytokines were explored using plasma samples. The only two which were detectable included CCL2/MCP-1 and IL-6, however, no significant differences were observed (Appendix A).

### 2.2. Serum-Derived Exosome Biomarker Levels in Concussed and Control Groups

Serum exosomes were prepared from a subset of participants in order to concentrate certain protein markers, normally expressed at low levels in crude serum, including Aβ42, and the p-tau proteins. There were no statistical differences in exosome Aβ42 concentrations between the concussed and control groups. The three isoforms of p-tau proteins, linked to neurodegenerative diseases (Alzheimers and ALS) were analyzed: p-tau181, p-tau217, and p-tau231 (Figure 2). Exosome-derived levels of p-tau181 were significantly higher in the concussed group compared to the healthy control group (* *p* < 0.05). Unlike p-tau181, the analysis of the concentrations of the isolated exosomes of p-tau217 and p-tau231 showed no differences between concussed and control groups. In terms of biomarker levels among participants, it was observed that certain individuals from the concussed group shared significantly high levels of more than one marker. For instance, one of the participants had higher exosome Aβ42 and p-tau231 levels and a large exosome size, while another athlete had higher levels of both Aβ42 and p-tau217.

### 2.3. Exosome Sizes in Concussed and Control Groups

The morphology of the exosomes was explored as a potential long-term biomarker of concussion. The analysis of exosome sizes (a random selection of 15–20 exosomes in each sample) revealed that the average diameters of exosome size (nm) were significantly higher (** *p* < 0.01) in players with concussion histories compared to the healthy control group (see Figure 3). The sizes of exosome samples isolated ranged from 22–39 nm (Figure 3).

### 2.4. Correlations among Biomarkers

A standard regression Pearson’s correlation analysis was performed between all the markers. Interestingly, there was a significant correlation between t-tau levels and exosome size (R^2^ = 0.2267, *p* = 0.03; n = 20). There was a positive correlation between Aβ42 and Nf-L (R^2^ = 0.2642, *p* = 0.02 (n = 19). Notably, there was a highly negative correlation between t-tau and RBP-4 (R^2^ = 0.3990, *p* = 0.003; n = 19), as well as RBP-4 and exosome sizes (R^2^ = 0.2955, *p* = 0.01; n = 20) (Figure 4).

## 3. Discussion

This study has provided significant new findings in terms of some of the key biomarkers found in the highly concussed group that could be linked chronically to long-term consequences including neurodegenerative diseases. We report significant differences in the levels of serum t-tau, RBP-4, and exosome tau-p181 in the group with a substantial history of concussion compared to the healthy control group. The average serum exosome sizes were larger in the concussed group compared to the healthy control group, indicating more cellular damage. These results could be used as a catalyst to re-evaluate the concussion protocols and long-term post-retirement outcomes in sports. These data provide a scientific basis for future research into the relationship between chronic biomarkers and development of neurodegenerative disease and indicate potential biomarkers that could be used for early diagnosis of concussion-based neurodegenerative diseases (CTE, AD, and ALS).

From our evaluation of the levels of t-tau in the serum of retired rugby players, we found that there was a significantly higher level of t-tau in the chronic concussion group compared to the control group. This finding supports a number of previous studies, including in American football players, where greater exposure to repetitive head injury over time was correlated with elevated later-life plasma t-tau concentrations [23]. Moreover, another study found higher blood concentrations of t-tau were associated with sport-related concussions at medical clearance [22]. In agreement with our findings, chronically elevated t-tau has been found in soldiers with a history of reported or medically documented concussion up to 18 months post-deployment, and elevated levels have also been observed at one-month post-severe TBI [53]. The effect on t-tau cannot be explained by the effects of aging [21]. The mean ages of the two participant groups were not significantly different. Recently, it was found that patients with ALS showed significantly higher levels of CSF t-tau, indicating that this protein may be a useful diagnostic biomarker of ALS [54].

Notably, the exosomal levels of p-tau181 were significantly higher in the group with a substantial history of concussion compared to the healthy control group. Devoto et al. [26] observed that concentrations of plasma p-tau181 in cohorts of patients with acute mTBI and concussed athletes were significantly high compared to controls. Furthermore, plasma p-tau181 levels were higher in retired contact sports players compared to the control group [27]. Previous studies indicated that p-tau181 was at the highest levels in patients with AD [55,56]. Moreover, plasma p-tau181 distinguished between non-AD and AD neurodegenerative disorders with accuracy [56]. During the symptomatic (prodromal and dementia) stages of AD, plasma p-tau181 increased significantly, while it was not elevated in non-AD [56]. Vorn et al. [57] observed that serum p-tau181 concentrations were increased over time following repetitive blasts among 34 male military personnel who participated in the breaching training program. This result showed a similar trend that brain exposure to external negative effects of trauma resulted in chronic elevation of p-tau181 levels suggesting this protein may serve as a promising biomarker of the long-term chronic consequences of brain injury. Recently, it was discovered that plasma p-tau181 was elevated in patients with ALS [58,59,60], suggesting that this protein may also be a novel marker to diagnose this disease.

Unlike p-tau181, the analysis of concentrations of both isolated exosomal p-tau217 and p-tau231 showed non-significant differences between concussed and control groups. Despite this finding, there was a tendency for an increase in the concentrations of p-tau217 and p-tau231 in some samples of retired rugby players compared to the control group. Prior studies found that p-tau217 discriminated AD from other neurodegenerative diseases, with significantly higher accuracy than established plasma- and MRI-based biomarkers [29]. Plasma p217 levels elevate early in the AD continuum, and therefore support a diagnosis of AD in persons [61]. Plasma p-tau217 levels correlate well with concurrent brain Alzheimer’s disease pathophysiology and with prospective cognitive performance [62]. It was also revealed that p-tau181, p-tau217, and p-tau231 increased early in the preclinical stages of the AD continuum [63].

This study found that there was no significant difference between the levels of SAA in the serum of the highly concussed rugby players group and the healthy control group. Our data are from the chronic time period, in contrast to the literature that measured acutely after TBI. Therefore, it can be concluded that SAA falls back to normal levels following TBI or concussion and the injury has no long-term effects on the levels of SAA in serum, as also reported in a mouse TBI study [64]. Thus, SAA may be a neuroinflammation-based diagnostic and prognostic biomarker for patients with TBI and not a chronic marker [64]. With regard to cytokines, again in contrast to acute concussion studies, this study did not find any statistical differences in IL-6 levels between people with a substantial history of concussion and control groups [46,47].

The use of Aβ42 as a biomarker of mild cognitive impairment and AD has been suggested in many studies [65]. There were no significant changes in Aβ42 in this present study, although there were some individuals with notably higher levels of Aβ42 in the serum of the high concussion compared to the healthy control group. Studies have found that the levels of Aβ42 increased in chronic concussion groups compared to the healthy control group [66]. Further studies are required to explore this further.

Vitamin A and other retinoids are involved in cellular regulatory processes in the nervous system [67]. Disrupted retinoic acid transport or signaling causes inflammatory responses, mitochondrial impairment, oxidative stress, cell differentiation, neurite outgrowth, and neurodegeneration [68]. Reduced retinoids may also negatively influence amyloid beta processing and directly inhibit the formation of amyloid fibrils in vivo [69]. This present study found the level of serum concentration of RBP-4 (retinoid transport mechanism), but not retinol, was lower in the highly concussed group compared to the healthy control group. Our findings were consistent with the literature that has shown that the serum levels of RBP-4 were dramatically reduced following TBI and were also associated with the relative risk and prognosis of ALS [70]. This suggests that both acute and chronic serum levels of this protein could be useful predictors for concussed patients with a brain injury [71] and offers a rationale for potential retinoid-based therapeutic interventions. Moreover, this present study observed a negative correlation between RBP-4 and t-tau, indicating the dual relevance to tauopathies, ALS, and CTE. Further retinoid pathway markers require exploration, including aldehyde dehydrogenases and CRABP proteins.

There was no significant difference between the serum levels of Nf-L in the highly concussed rugby players group and the healthy control group. Previous studies demonstrated that the vitamin A levels in the cerebral cortex were not affected by TBI, again an acute injury [72]. In a boxing study, it should be noted that Nf-L levels were found to decrease after three months of rest but were yet to return to baseline levels [39]. These findings were also supported by another study in American football players, as levels of Nf-L were observed to increase over the course of a season in American football players [73], suggesting that Nf-L could be a marker of acute concussion or repeated subconcussive head impacts during a career, but not long-term. Even though this study failed to find any significant differences in Nf-L serum levels between the two retired groups, an interesting modest positive correlation between exosome Aβ42 and plasma Nf-L was observed. This requires further investigation. Notably, since acute concussion diagnosis currently relies only on physical examination results, the Abbott diagnostics of GFAP and UCH-L1 from venous blood, and clinical behavioral observations [74], discovering quantitative-based measurements such as serum/plasma biomarkers and exosome dimensions may contribute to improved detection and understanding of the long-term effects of multiple rugby career concussions in retirement and better prophylactic care provision.

In conclusion, this present study provided new evidence linking multiple concussions in rugby to a long-term combination of elevated serum tau and p181-tau and a reduced retinoid biomarker, RBP-4, both effects previously associated with chronic tauopathies, including ALS and CTE. Moreover, exosomes could be used to concentrate biomarkers to facilitate their detection as a novel chronic diagnostic tool for neurodegenerative tauopathies in individuals who have experienced multiple concussions [75], supported by the recently reported increase in mean exosome diameter in patients with ALS [76]. The current finding of a positive correlation between exosome size and t-tau, as well as a negative correlation between exosome size and RBP-4, provides a rationale for exploring the prophylactic clinical use for concussion based on a new class of retinoid receptor modulators (RAR-Ms), exemplified by Ellorarxine, currently MHRA approved to proceed to clinical trials for ALS (www.nevrargenics.com).

## 4. Materials and Methods

### 4.1. Study Design and Setting

The current research analyzed a range of selected biomarkers in retired rugby players with a history of concussion and non-contact sports controls from the UK Rugby Health Project [6]. The UK Rugby Health Project was initiated in 2016 as an extension to the inaugural New Zealand Rugby Health Project [77] and the methods and results so far have been published elsewhere [1,6,78,79].

### 4.2. Study Participants

Former male rugby players and non-contact sport athletes took part in the study and were recruited from September 2016 to December 2018 using past player/athlete associations, printed and televised media reports, word of mouth, and social media. Participants in the current study were those who had attended an in-person clinical appointment and provided a fasted blood sample. The concussed group in this study were retired rugby players who reported 5 or more diagnosed concussions during their sporting career (n = 30), and the control group were both retired rugby players and retired non-contact sports participants with no reported concussions (n = 26). The number of replicate analyses reflected the limited availability of sample volumes for randomly selected age-matched groups of participants (n = 10–23), which reflects a limitation of this present study. No significant differences were seen in participant ages or mean years at and since retirement from sport (*p* > 0.05). The severity of the individual concussions was not monitored, and only the group with over 5 concussions was explored in this initial study to explore the extremes in the first instance. Further studies will follow to meet these limitations.

A general health questionnaire was used to gather information regarding engagement in rugby, injuries, including concussions, present health and well-being, and height and weight measurements [1,6]. The questionnaire was accessible online from September 2016 to December 2018.

### 4.3. Biomarker Assays

This project investigated concentration levels of serum (t-tau, RBP-4, SAA, and retinol), plasma Nf-L and cytokines, and serum-derived exosomes (Aβ42, p-tau181, p-tau217, and p-tau231) for the highly concussed cohort and the non-concussed UK athletes [77,78,79,80,81]. Additionally, serum-derived exosome sizes between the same two groups were measured.

#### 4.3.1. Exosomes Preparation

To isolate exosomes from serum, the Total Exosome Isolation reagent was used (Thermo Fisher Scientific, Dartford, UK, catalog no: 4478360). Blood samples were allowed to clot and centrifuged at 2000× *g* for 5 min to remove cells and debris. Once completed, we transferred the supernatant containing the clarified serum to a new tube without disturbing the pellet and these were snap frozen and stored at −80 °C (2016–2017). Anonymous labeling of the tubes ensured data protection. We took the required volume of clarified serum for each experiment, and then added the total exosome isolation reagent in a 5:1 ratio of serum–reagent. Once added, the samples were allowed to incubate for 30 min at 2–8 °C. Next, the samples were centrifuged again at 10,000× *g* for 10 min at room temperature. Finally, the supernatant was removed and discarded, and the pellet was reconstituted in the desired volume of PBS for the subsequent assay.

#### 4.3.2. Exosome Size Measurement

Total Exosome Isolation Reagent supplied by (Thermo Fisher Scientific, catalog no: 4478360), was used to isolate the exosomes as in Section 4.3.1, followed by filtration through a sterile 0.2 μm filter to remove any remaining large debris material, prior to preparation for TEM analysis. The identity of exosomes was confirmed using a CD-63 antibody marker (Abcam, Cambridge, UK Catalog number EPR21151, unpublished). To measure exosome sizes, the isolated exosome samples were mounted on microscope slides using an unfixed sample method. In this assay, 10 uL of each sample was washed for two minutes with MQ water, then 10 uL of 1% uranyl acetate was added. For this method, a 50,000 magnification was used. After capturing the exosome images, they were saved in a file and then measured using image-J software (Fiji computer software, version 2.15.1).

### 4.4. ELISA Assays

Assays performed using serum followed manufacturer instructions for t-tau (Thermo Fisher Scientific, catalog no: KHB0041), RBP-4 (Thermo Fisher Scientific, catalog no: BMS2199), SAA (Thermo Fisher Scientific, catalog no: EHSAA1), Nf-L (Bioassay Technology Laboratory, Shanghai, China, catalog no: E4467Hu), and retinol (Abbexa, Cambridge, UK, catalog no: abx150383). Assays performed using serum exosomes followed manufacturer instructions for Aβ42 (Abcam, Cambridge, UK, catalog no: ab289832), p-tau181 (Thermo Fisher Scientific, catalog no: KHO0631), p-tau217 (MyBioSource, San Diego, CA, USA, catalog no: MBS1608795), and p-tau231 (Thermo Fisher Scientific, catalog no: KHB8051). The proinflammatory chemokine CCL2/MCP-1 and cytokine IL-6 were assessed in plasma using a Human Luminex^®^ Discovery Multiplex Assay (R&D Systems, Minneapolis, MN, USA, catalog no: LXSAHM), according to manufacturer directions, and read on a Luminex^®^ MAGPIX^®^ analyser (R&D Systems).

### 4.5. Statistical Analyses

Data were processed in Microsoft Excel 2023 and GraphPad Prism software version 10 was used for all statistical analyses, including the calculation of serum and exosome concentrations, means, standard deviations (SD), coefficient of determination (R^2^), and *p* values (where * significant *p* < 0.05; ** highly significant *p* < 0.01; n.s. denotes non-significant). The test carried out was a Mann–Whitney U test. All quantitative data are expressed as mean values ± SD of the mean for n = 10–23 individual cases. To visualize the exosomes, electron microscopy was used, while exosome images were analyzed using Image J software (Fiji computer software, version 2.15.1), and measured blind to the observer.

## 5. Conclusions

Our findings propose that a selection of specific biomarkers could be measured in the serum and serum exosomes of participants who have previously suffered multiple concussions in their career that could be predictive, years later, of negative outcomes such as neurodegenerative disorders. Notably, this study found significant differences in the levels of serum t-tau, RBP-4, and exosome tau-p181 in the group with a substantial history of concussions compared to the healthy control group. It also found that the average serum exosome sizes were larger in the highly concussed group compared to the healthy control group. These results could be used as a catalyst to re-evaluate concussion protocols and long-term post-retirement outcomes in sports.

## Figures and Tables

**Figure 1 ijms-25-07811-f001:**
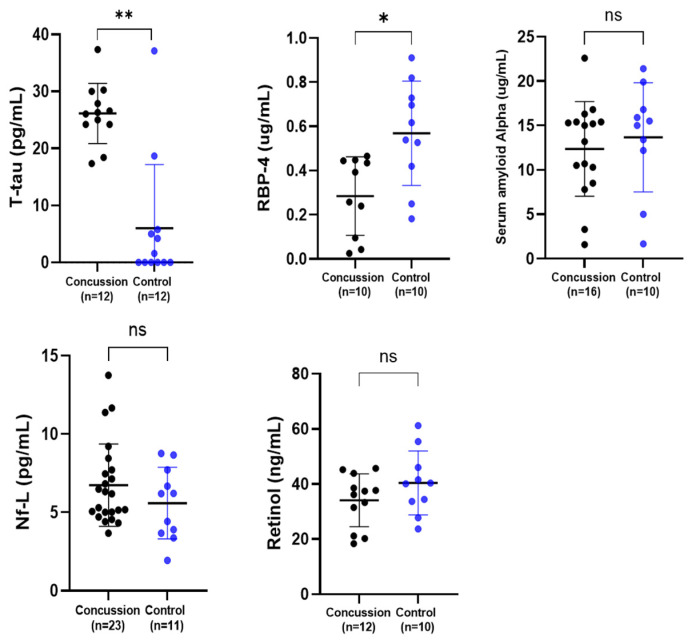
Comparison of the serum concentration of t-tau, RBP-4, SAA, Nf-L, and retinol in a concussion vs. control group. Each dot represents an individual data point with the bars representing the median and the range. Blue dots = the control group, and black dots = the concussion group. * *p* < 0.05; ** *p* < 0.01, ns not significant.

**Figure 2 ijms-25-07811-f002:**
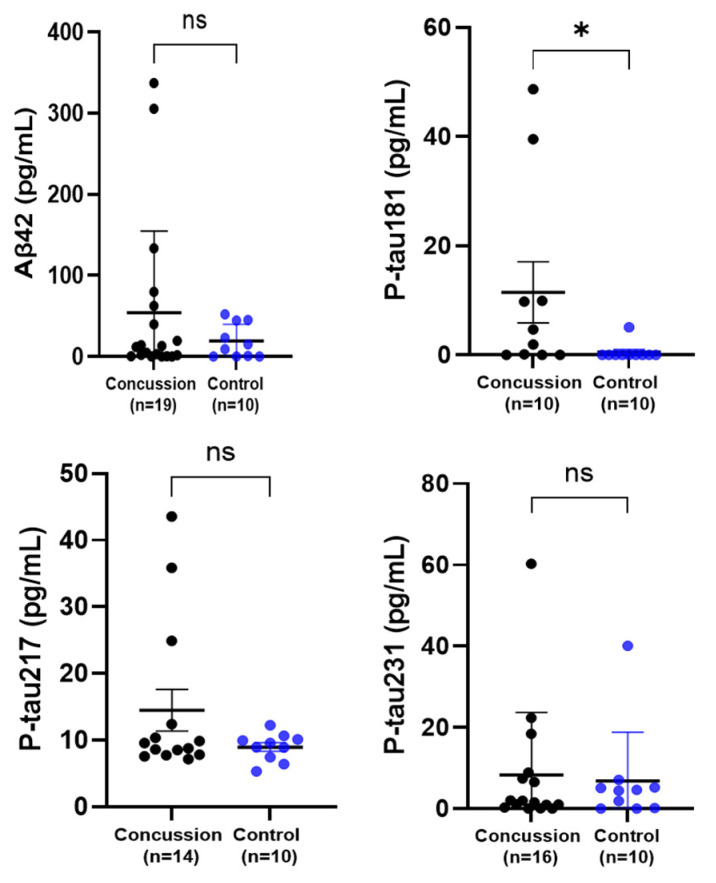
Comparison of the exosome concentrations of Aβ42, p-tau181, p-tau217, and p-tau231 in a concussion vs. control group. Each dot represents an individual data point with the bars representing the median and the range. Blue dots = the control group, and black dots = the concussion group. * *p* < 0.05, ns not significant.

**Figure 3 ijms-25-07811-f003:**
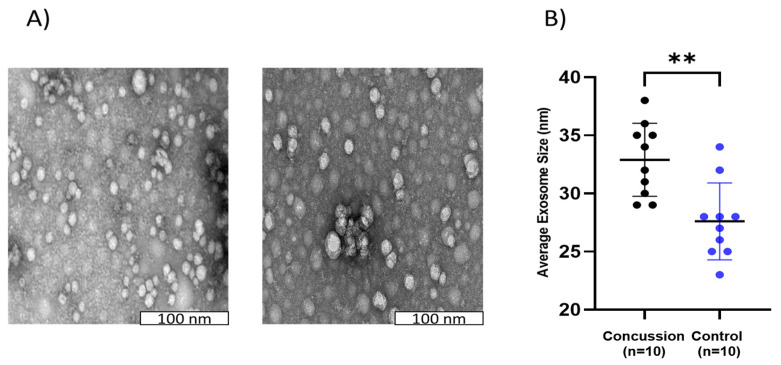
Comparison of the exosome size in a concussion vs. control group. A. Images of exosome sizes in the concussed group. Electron microscopy was used to visualize the structure of the exosome size (**A**). The scale bar was 100 nm. B. A significant difference was observed between the concussion group compared to the control group. ** *p* < 0.01 (**B**). Each dot represents an individual data point with the bars representing the median and the range. Blue dots = the control group, and black dots = the concussion group.

**Figure 4 ijms-25-07811-f004:**
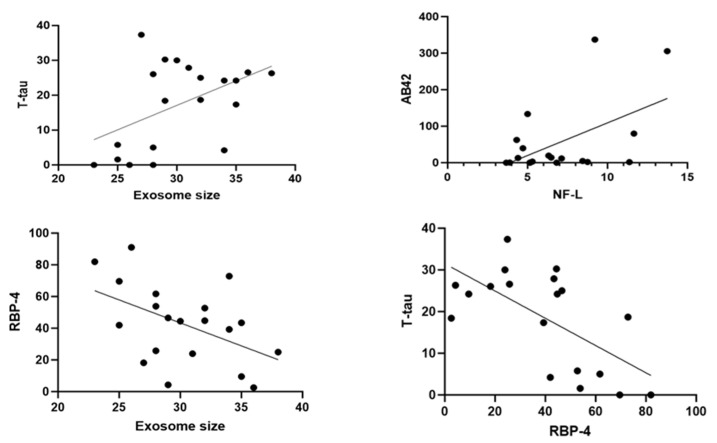
Scatter plots to highlight the correlation between biomarkers. Dots are individual data points for the concentration of the substances specified or exosome sizes on graphs of specific serum samples (n = 20 or more).

**Table 1 ijms-25-07811-t001:** Descriptive characteristics.

	Concussed Group N = 30	Control Group N = 26
Mean ages	41.84 ± 6.44	45.75 ± 11.52
Mean ages at retirement	33.55 ± 5.82	35.47 ± 10.12
Mean years since retirement from the sport	7.76 ± 6.69	6.75 ± 4.64
Playing position	5 prop, 4 hooker, 1 forward, 1 s row, 1 fly half, 6 center, 3 wing, 2 backward, 1, openside flanker, 1 blindside flanker, 1 number 8, 2 lock, 1 standoff.	1 blindside flanker, 2 backwards, 1 wing, 1 number 8, 1 prop, 1 standoff, 20 non-athletes.
Mean weight	91.5 ± 31.2	88.2 ± 34.4
Mean height	183.0 ± 7.4	179.4 ± 6.6
Rugby league (RL) or union (RU)	12 (RL), 17 (RU)	5 (RU), 2 (RL), 20 (N/A)

## Data Availability

Data is contained within the article and Appendix A.

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
