# Peer review of "Concussion-Related Biomarker Variations in Retired Rugby Players and Implications for Neurodegenerative Disease Risk: The UK Rugby Health Study"

_ijms, 2024, doi:10.3390/ijms25147811_

Round 1

Reviewer 1 Report

Comments and Suggestions for Authors

The authors present an interesting study examining the levels of markers associated with concussion/neurodegenerative disorders in cohorts of retired athletes, namely rugby players and non-contact sport athletes. Briefly, utilising serum samples collected from the participants, the authors profile the contents for reactive proteins, cytokines, chemokines, and exosomes in an effort to correlate incidence of concussion with changes in serum profile. Of note, the ‘concussion’ cohort demonstrated slightly larger exosomes as compared to the ‘control’ group, with interesting correlations found between exosome size and level of inflammatory mediator. Overall this was an interesting read, but in performing my review I made a number of observations. The following should be considered by the author when preparing a suitable revision:

1.      Why were rugby players included in the control group? Would it not have been more straightforward to go with purely non-athletes as those rugby players may have experienced concussion in training, matches, etc that went unreported.

2.      With access to 30 and 26 individuals in both the concussed and control groups respectively, why was not every person examined in the biomarker assays? Moreover, why is there a non-equal number in certain instances i.e. serum amyloid alpha Figure 1?

3.      Were the samples collected and processed fresh, or were they part of a biobank from a previous study? The age of the samples may influence the levels measured, as some are reported as being undetectable.

4.      Why wasn’t an instrument for gaining average sizes in a sample used for the measurement of the exosomes? An instrument that uses dynamic light scattering or similar to get a greater average/representative value in the sample.

5.      Were the exosomes every profiled to determine they are exosomes or some other form of extracellular vesicle?

6.      More details on the method of exosome preparation are required. For example, what reagent exactly was used for the isolation? Company and catalogue number should be given. Also, were the samples screened to examine the profile of the sample overall? A much more mixed population would be expected based on this method, and this raises questions of the exosome images presented.

7.      In performing the analyses, did the authors consider analysing the data with respect to number of concussions sustained by the individual, the severity of each concussion, and/or whether the appropriate level of rehabilitation was adhered to with respect to serum profile?

Author Response

The authors present an interesting study examining the levels of markers associated with concussion/neurodegenerative disorders in cohorts of retired athletes, namely rugby players and non-contact sport athletes. Briefly, utilising serum samples collected from the participants, the authors profile the contents for reactive proteins, cytokines, chemokines, and exosomes in an effort to correlate incidence of concussion with changes in serum profile. Of note, the ‘concussion’ cohort demonstrated slightly larger exosomes as compared to the ‘control’ group, with interesting correlations found between exosome size and level of inflammatory mediator. Overall this was an interesting read, but in performing my review I made a number of observations. The following should be considered by the author when preparing a suitable revision:

Thank you for your review and positive comments. We would like to emphasise that the key results reported in this study relate to the significant raised levels of t-tau and exosome tau-p181, together with reduction RBP-4 retinoid markers, as well as the larger exosomes. The strong correlations between t-tau and retinoid marker and exosome size are most relevant to the field and potential use as chronic biomarkers and interventions as we stated in manuscript. The cytokine data is less significant, although due to the low levels, it was challenging to conclude too much, hence inclusion in supplementary data sets.

  1. Why were rugby players included in the control group? Would it not have been more straightforward to go with purely non-athletes as those rugby players may have experienced concussion in training, matches, etc that went unreported.

The control group contained rugby and non-contact sport individuals with no reported concussions during their career. We tried to ensure that this was the case, as strictly as we could.

  1. With access to 30 and 26 individuals in both the concussed and control groups respectively, why was not every person examined in the biomarker assays? Moreover, why is there a non-equal number in certain instances i.e. serum amyloid alpha Figure 1?

The numbers reported were variable, reflecting the availability of sample volumes for each participant. The preparation of exosomes required a large volume, to ensure detectability of some of the markers (B-amyloid, p-tau proteins). We ensured that there was at least n=10 per group to deliver statistical power.

  1. Were the samples collected and processed fresh, or were they part of a biobank from a previous study? The age of the samples may influence the levels measured, as some are reported as being undetectable.

The samples were collected and processed fresh before freezing and storage at -80°C seven years ago. Yes, the age of the sample may influence the stability of the markers, but the individual assays were performed in parallel (concussed and control samples) on freshly defrosted samples to ensure consistency in comparisons. All samples were kept in the same storage conditions until assaying.

  1. Why wasn’t an instrument for gaining average sizes in a sample used for the measurement of the exosomes? An instrument that uses dynamic light scattering or similar to get a greater average/representative value in the sample.

We agree with reviewer, but such a device was not available to us at the time. However, a large number of samples were manually analysed, and importantly, samples were collected and analysed blind to researcher. Such a device will be sought in the future to simplify the analysis.

  1. Were the exosomes every profiled to determine they are exosomes or some other form of extracellular vesicle?

The identity of the exosomes was achieved using a suitable marker (eg. CD-63, data not shown), further markers, CD9 and CD81 will be explored going forward)

  1. More details on the method of exosome preparation are required. For example, what reagent exactly was used for the isolation? Company and catalogue number should be given. Also, were the samples screened to examine the profile of the sample overall? A much more mixed population would be expected based on this method, and this raises questions of the exosome images presented.

Total Exosome Isolation Reagent supplied by (Thermo Fisher Scientific, catalog no: 4478360), was used followed by filtration through a sterile 0.2 µm filter to remove any remaining large debris material, prior to preparation for TEM analysis. ID of exosomes was achieved suing a CD-63 antibody marker (data not shown). To examine the profile of the exosome samples, a morphological examination method was performed using Transmission Electron Microscopy (TEM). To measure exosome sizes, the isolated exosome samples were mounted on microscope slides using an unfixed sample method. In this assay, 10 uL of each sample was washed for two minutes with MQ water then 10 ul 1% uranyl acetate was added. For this method, 50,000 magnification was used, and the wavelength was measured at ~100 nm to achieve high resolution. After capturing the exosome images, they were saved in a file and then measured using image J software. The sizes of exosome samples ranged from 22 - 39 nm.

  1. In performing the analyses, did the authors consider analysing the data with respect to number of concussions sustained by the individual, the severity of each concussion, and/or whether the appropriate level of rehabilitation was adhered to with respect to serum profile?

Yes, we did consider this suggestion, but for this first study, we decided to look at the extremes ie. > 5 concussions Vs controls. Also details on severity of concussions were not available. Future work will explore such suggestions and there is indeed an urgent need for funding to perform a follow up study to evaluate the status of the participants in the present day, as a number had worrying multiple biomarker profiles irregularities 7 years ago.

Reviewer 2 Report

Comments and Suggestions for Authors

The study explores an important topic - the long-term effects of concussions on biomarkers related to neurodegenerative diseases in retired rugby players. The findings provide new insights that could help better understand the chronic consequences of repeated concussions.

  • The sample sizes for some of the biomarker analyses are relatively small (n=10-23). Increasing the sample size would provide more confidence in the findings, especially for markers that showed non-significant differences but an increasing trend in the concussed group.
  • More details could be provided on the inclusion/exclusion criteria for selecting participants, to ensure no other potential confounding factors between groups.
  • It would be informative to classify the concussed group into further sub-groups based on the number and severity of concussions, and years since last concussion, to examine any dose-response or temporal relationships with the biomarkers.
  • The discussion can further elaborate on the potential mechanisms linking the altered biomarkers with chronic neurodegeneration, and highlight the most promising biomarker candidates for future validation.
  • Proofreading is needed to fix grammatical and typographical errors in some parts of the manuscript.

Overall, this is a valuable study that identifies novel chronic biomarkers of neurodegeneration in retired athletes with a history of repeated concussions. With some additional data and revisions, it can make a significant contribution to our understanding of the long-term risks of concussions. The biomarkers identified hold promise as potential screening tools for neurodegeneration in high-risk populations.

Author Response

The study explores an important topic - the long-term effects of concussions on biomarkers related to neurodegenerative diseases in retired rugby players. The findings provide new insights that could help better understand the chronic consequences of repeated concussions.

We thank the reviewer for his/her kind comments, and we agree about the importance of the study.

  • The sample sizes for some of the biomarker analyses are relatively small (n=10-23). Increasing the sample size would provide more confidence in the findings, especially for markers that showed non-significant differences but an increasing trend in the concussed group.

We agree with the reviewer. The numbers reported were variable and reflected the availability of sufficient volumes for each participant. The preparation of exosomes required a large volume, to ensure detectability of some of the key biomarkers (B-amyloid, p-tau proteins). We ensured that there were at least n=10 per group to deliver statistical power.

  • More details could be provided on the inclusion/exclusion criteria for selecting participants, to ensure no other potential confounding factors between groups.

Samples tested were selected randomly and purely dependent on sufficient serum sample volumes for exosome preparation and no other factors. Age of participants and time after retirement in the two test groups were not significantly different. The control participants were selected with as strict criteria as possible relating to concussions.

  • It would be informative to classify the concussed group into further sub-groups based on the number and severity of concussions, and years since last concussion, to examine any dose-response or temporal relationships with the biomarkers.

Yes, we did consider this suggestion, but for this first study, we decided to look at the two extremes ie. > 5 concussions Vs controls. Also details on severity of concussions were not available. Future work will explore such suggestions and there is an urgent need for funding to perform a follow up study to evaluate the status of the participants in the present day, as a number had multiple biomarker profiles irregularities 7 years ago.

  • The discussion can further elaborate on the potential mechanisms linking the altered biomarkers with chronic neurodegeneration, and highlight the most promising biomarker candidates for future validation.

We have developed this a little more in the discussion. t-tau, exosome tau-p181 and RBP-4 profiles look the most promising.

In conclusion, this present study provided new evidence linking multiple concussions in rugby to a long-term combination of elevated serum tau and p181-tau and a reduced retinoid biomarker, both effects previously associated with chronic tauopathies, including ALS and CTE.”

  • Proofreading is needed to fix grammatical and typographical errors in some parts of the manuscript.

Proofreading was performed to correct any errors.

Overall, this is a valuable study that identifies novel chronic biomarkers of neurodegeneration in retired athletes with a history of repeated concussions. With some additional data and revisions, it can make a significant contribution to our understanding of the long-term risks of concussions. The biomarkers identified hold promise as potential screening tools for neurodegeneration in high-risk populations.

We thank the reviewer for his/her kind comments, and we agree about the importance of the study.

Round 2

Reviewer 1 Report

Comments and Suggestions for Authors

The authors have suitably addressed my comments for the most part. I do believe that on some points where the limitations are agreed upon this could be added to the text to acknowledge such.   

Author Response

Thanks you to the reviewers for their useful comments.

This paragraph was modified to meet this point regarding limitations of the study.

The number of replicate analyses reflected the limited availability of sample volumes for randomly selected age-matched groups of participants (n=10-23), which reflects a limitation of this present study. No significant differences were seen in participant ages, mean years at and since retirement from sport (P>0.05). The severity of the individual concussions was not monitored, and only the group with over 5 concussions were explored in this initial study, to explore the extremes in the first instance. Further studies will follow to meet these limitations.